# Emerging Functional Imaging Biomarkers of Tumour Responses to Radiotherapy

**DOI:** 10.3390/cancers11020131

**Published:** 2019-01-23

**Authors:** Alan Campbell, Laura M. Davis, Sophie K. Wilkinson, Richard L. Hesketh

**Affiliations:** 1Department of Clinical Radiology, University College London Hospital, 235 Euston Road, London NW1 2BU, UK; alan.campbell3@nhs.net (A.C.); lauramaydavis@nhs.net (L.M.D.); sophiekristina.wilkinson@nhs.net (S.K.W.); 2Cancer Research UK Cambridge Institute, Li Ka Shing Centre, Robinson Way, Cambridge CB2 0RE, UK

**Keywords:** radiotherapy, functional imaging, magnetic resonance imaging, positron emission tomography, hyperpolarised ^13^C, diffusion-weighted imaging, treatment response

## Abstract

Tumour responses to radiotherapy are currently primarily assessed by changes in size. Imaging permits non-invasive, whole-body assessment of tumour burden and guides treatment options for most tumours. However, in most tumours, changes in size are slow to manifest and can sometimes be difficult to interpret or misleading, potentially leading to prolonged durations of ineffective treatment and delays in changing therapy. Functional imaging techniques that monitor biological processes have the potential to detect tumour responses to treatment earlier and refine treatment options based on tumour biology rather than solely on size and staging. By considering the biological effects of radiotherapy, this review focusses on emerging functional imaging techniques with the potential to augment morphological imaging and serve as biomarkers of early response to radiotherapy.

## 1. Introduction

The introduction of cross-sectional anatomical imaging using computed tomography (CT) and magnetic resonance imaging (MRI) in the 1970s revolutionised clinical oncology by permitting non-invasive determination of tumour burden that is essential for diagnosis and staging and assessing treatment responses. To standardise characterisation of tumour responses that were frequently incomplete and heterogeneous and for comparison between clinical trials the World Health Organisation defined the first response criteria in 1979 [1]. Although subsequently refined the criteria used today, most commonly in the form of the Response Evaluation Criteria in Solid Tumours (RECIST) [2,3], are recognisably similar (Table 1). Measurements of tumour burden are often an excellent determinant of disease progression or response but changes usually manifest themselves slowly and can sometimes be misleading [4]. In the forty years since the introduction of size response criteria, functional imaging techniques have emerged that are capable of reporting many aspects of tumour biology. In response to therapy, biochemical changes precede anatomical changes, sometimes by many months [5]. Earlier determination of response to treatment would facilitate modification of treatment before significant disease progression and reduce the physical, psychological and financial costs of ineffective or unnecessary therapy. Of the fourteen million people diagnosed with cancer worldwide every year, more than half receive radiation therapy [6]. Identification of radio-resistant tumour regions at a pre- or early therapy stage could be used for localised or global dose escalation or initiation of concomitant chemotherapy. Following treatment, functional imaging is a potentially powerful tool where the conclusions of morphological imaging can be limited, for example, differentiation of radiation necrosis and residual disease in the brain.

In this review we will focus on emerging functional imaging techniques that exploit the biological changes in tumours following radiation therapy and have the potential to improve the early detection of treatment response. Many of the functional imaging techniques discussed can also be applied to delivering intensity-modulated or stereotactic body radiation therapy using increasingly sophisticated methods, a subject that is beyond the scope of this review but has been extensively reviewed elsewhere [7,8]. 

## 2. Biological Effects of Radiation

Ionising radiation refers to particles that have sufficient energy to release electrons from an atom. The most significant biological target of ionising radiation is DNA that can be ionised directly or indirectly by free radicals (e.g., hydroxyl (^•^OH), superoxide (O_2_^−^) and hydrogen peroxide (H_2_O_2_)) produced by ionisation of adjacent molecules [9,10]. DNA ionisation can result in damage to any part of the molecule. Base damage and single-stranded breaks occur frequently but efficient repair mechanisms limit the biological effect. In contrast, double-stranded breaks and DNA crosslinking are less frequent events but are also less likely to be repaired effectively resulting in either genomic mutation or repair failure and subsequent cell death. Radiation-induced cell death can result from activation of cellular senescence or apoptosis, the latter predominantly via the intrinsic pathway [11]. However, particularly in tumour cells where cell cycle checkpoint controls, DNA repair and apoptotic pathways are frequently perturbed, cell death predominantly occurs from mitotic catastrophe, a result of premature induction of mitosis before S and G_2_ phase completion that ultimately results in cell necrosis [9,12]. 

Radiobiological effects are dependent on external factors such as the dose and type of radiation used. For example, protons and alpha particles have a high linear energy transfer and are more likely to induce complex DNA damage with a higher probability of lethality [13]. Additionally, biological variables result in heterogeneous radiation sensitivity between and within tissues. Hypoxia and low rates of proliferation tend to promote radio-resistance and cancer stem cells may be more resistant than the bulk of tumour cells [14]. 

The effects of radiation on tumours are not limited to cell death with virtually every aspect of the tumour microenvironment responding to the insult. In the acute phase necrosis and vascular disruption leads to hypoperfusion, oedema and an inflammatory response that begins in the first few hours following acute radiation injury. Chronic activation of the inflammatory response results in dysregulated tissue remodelling characterised by decreased vascularity and fibrosis [14]. 

## 3. Imaging Apoptosis and Necrosis

Accurate determination of cell death would find application in a wide range of conditions including stroke, myocardial infarction and cancer. Several probes have been designed to assess biochemical events that occur during cell death. Phosphatidylserine, an anionic phospholipid and a major component of the inner leaflet of the cell membrane, is externalised by stressed or dying cells and is a target for phagocytosis [15,16]. Annexin-V binds to externalised phosphatidylserine with low nanomolar affinity and has been radiolabelled with ^18^F and ^99m^Tc for positron emission tomography (PET) and single photon emission computed tomography (SPECT) imaging, respectively [17]. ^99m^Tc-annexin-V has progressed to clinical trials, where increases in labelling of 20–30% in the first 72 h following chemotherapy or radiotherapy were associated with treatment response in lung, breast, lymphoma and head and neck cancers [18]. Unfortunately, annexin-V is limited by slow pharmacokinetics and high levels of non-specific binding, particularly to the abdominal organs [17]. An alternative is using the C2A domain of synaptotagmin-I which binds to anionic phospholipids and has been labelled with ^99m^Tc and ^111^In for SPECT [19,20], ^18^F for PET [21] and gadolinium chelates for magnetic resonance imaging (MRI) in preclinical in vivo studies [22]. To improve the biodistribution, simplify labelling and improve pharmacokinetics (which are often slow when using peptide-based tracers), a smaller modified C2A protein (C2Am) has been labelled with ^99m^Tc for SPECT (Figure 1) [20] and a near-infrared fluorophore for multispectral optoacoustic tomography (MSOT) [23], demonstrating high sensitivity and specificity for cell death. Tumour uptake of ^99m^Tc-duramycin which binds to phosphatidylethanolamine, another phospholipid externalised during cell death, has demonstrated improved sensitivity in detecting early treatment response compared to ^18^F-FDG in preclinical studies [24].

Several small-molecule imaging probes that detect cell or mitochondrial membrane depolarisation and/or acidification of apoptotic cells have also been developed. In patients with intracranial tumours, the change in ^18^F-ML-10 uptake from before to 48 h after CyberKnife stereotactic radiotherapy correlated with the decrease in tumour volume measured at 2–4 months after treatment [25]. Similar correlations have been made in patients with brain metastases imaged before and nine days after whole-brain radiotherapy [26]. 

Cell membrane changes are not specific to apoptosis and increased binding and uptake are also seen in autophagy, necroptosis and necrosis. Several PET radiotracers have been designed to detect cleaved caspase 3 and 7, components of the final common pathway of apoptosis that have greater specificity for apoptosis. Of these, ^18^F-ICMT-11 has recently been used in breast and lung cancer patients, although low tumour uptake explained by low cleaved caspase 3 expression before and after treatment limited the conclusions [27].

## 4. Imaging Changes in Vasculature

Radiation therapy results in acute endothelial cell dysfunction, apoptosis and disruption of blood vessels. Above doses of 8–10 Gy endothelial cell apoptosis is induced by activation of the acid sphingomyelinase (ASMase)/ceramide signalling pathway [28,29,30]. Therefore, activation of this pathway does not occur with the lower doses delivered in fractionated radiotherapy, only with the higher single doses delivered with stereotactic radiotherapy [31]. Capillaries increase in permeability and become thrombosed due to platelet aggregation and microthrombus formation with subsequent hypoperfusion causing further tumour necrosis [12,32]. This suggests that imaging changes in perfusion have potential for early detection of tumour responses to radiotherapy. 

### 4.1. Dynamic Contrast-Enhanced (DCE) CT

DCE-CT following an intravenous bolus of iodinated contrast agent is a highly reproducible imaging technique that permits relatively simple absolute quantification of blood flow, blood volume, permeability–surface area product, mean transit time and extravascular volume [33]. Correlation of DCE-CT metrics with histological determination of microvessel density and vascular endothelial growth factor (VEGF) expression has been possible in some studies [34,35]. Reductions in blood flow, blood volume, mean transit time and permeability–surface area product have been demonstrated in patients with rectal cancer, head and neck cancer and brain metastases following radiotherapy ± chemotherapy [35,36,37,38,39,40]. However, in patients with cervical cancer increases in tumour blood volume were observed three weeks into chemoradiotherapy which were predictive of complete metabolic response at three months [41]. The conflicting findings may reflect heterogeneity between tumour types and responses to treatment but also differences in timing of the post-treatment study.

### 4.2. Perfusion MRI

Following injection, paramagnetic contrast agents (typically low-molecular-weight gadolinium (Gd^3+^) chelates) are distributed via the blood and diffuse freely into the interstitial space but do not cross the cell membrane. Paramagnetic contrast agents cause magnetic field inhomogeneities that reduce the *T*_1_, *T*_2_ and *T*_2_* relaxation times of nearby protons resulting in temporal changes in MR signal intensity and can provide information on the concentration of the injected contrast agent, microvessel density, perfusion and vessel permeability [42,43,44]. The most commonly used techniques are DCE-MRI and dynamic susceptibility contrast MRI, which exploit the *T*_1_ and *T*_2_* effects of paramagnetic contrast agents, respectively. In addition to subjective visual analysis of the rate, total amount and decrease (washout) of contrast enhancement in lesions, semi-quantitative and quantitative parameters can be derived similar to those of DCE-CT, although the post-processing is complicated by a nonlinear relationship between contrast agent concentration and change in signal intensity [33]. The use of an exogenous contrast agent can be avoided by using arterial spin labelling (ASL) in which blood water protons are magnetically labelled. This suffers from low temporal and spatial resolution and low signal-to-noise ratio but has greatly reduced post-processing requirements when compared to imaging exogenous contrast agents [45].

In patients with cervical cancer high contrast enhancement before and in the first few weeks after the initiation of chemoradiotherapy is a better predictor of response than tumour volume measurements [46,47,48]. Other semi-quantitative and quantitative measures, particularly higher rates of *K*^trans^ (the volume transfer constant between plasma and the extravascular extracellular space) and plasma flow, have also been shown to be predictive of response [47,49,50]. Similar results have been obtained for rectal cancer and head and neck cancer where a high *K*^trans^ before chemoradiotherapy and a large decrease or low *K*^trans^ after therapy are generally associated with good response [51,52]. In high-grade gliomas and cerebral metastases, reductions in *K*^trans^ and changes in tumour blood volume and flow (from DSC-MRI and ASL) have detected response to stereotactic radiosurgery or whole-brain irradiation as early as one week after treatment [53,54,55]. Correlations between high perfusion on DCE-MRI and radio-sensitivity have frequently been attributed to decreased hypoxia in well-perfused tumours [46]. However, it has also been reported that higher microvessel density and increased angiogenesis correlates with greater metastatic potential and poorer outcome [56]. 

In addition to prognostication, another potential application of perfusion MRI is the differentiation between radiation necrosis and recurrence in high-grade glioma, which often appear similar using conventional contrast-enhanced MRI [57]. A meta-analysis concluded that sensitivity and specificity for tumour recurrence was 90% and 88%, respectively, using DSC-MRI and 89% and 85% with DCE-MRI [58]. Initial studies using ASL have also demonstrated its ability to differentiate disease recurrence from radiation necrosis with a high degree of accuracy (Figure 2) [59,60,61,62]. 

### 4.3. Ultrasound and Optical Imaging

Radiation-induced changes in vasculature can also be imaged using dynamic contrast-enhanced ultrasound (CEUS). Low solubility, gas-containing microbubbles have different acoustic properties and can be used to image microvascular density and perfusion. This has been used to predict response of a number of different cancer types to chemotherapy [63,64,65], while decreased vascular density following radiation therapy has been used as an early marker of response in preclinical tumour models [66,67]. Furthermore, using antibodies conjugated to the surface of microbubbles could permit tumour targeting. Microbubbles targeting the angiogenesis regulators α_v_β_3_ integrin and ICAM-1 that are upregulated in response to radiotherapy were increased in a rat prostate tumour model so treated [68]. 

Optical coherence tomography (OCT) is a non-invasive imaging technique that can produce 3D in vivo images at a resolution of a few micrometres by measuring the interference pattern of back-scattered light [69]. Although OCT has unrivalled spatial resolution, the scattering of light within biological tissues limits the imaging depth to a few millimetres. OCT is well established for high-resolution 3D retinal imaging and, more recently, functional imaging of the microvasculature of tumours has been demonstrated using speckle variance OCT [70]. In pancreatic human tumour xenografts irradiated with ≥10 Gy, the vascular volume density decreased by 26% just 30 min post-radiotherapy. Early changes were predominantly seen in small vessels <30 μm in diameter and were transient, potentially indicating rapid microthrombus formation following radiotherapy [71]. Maximal reductions in vascular volume density were seen after 2–4 weeks, depending on delivered dose, and preceded reductions in tumour volume by several weeks [70].

MSOT uses an ultrasound transducer to measure acoustic waves generated in response to localised thermoelastic expansion of tissue induced by pulses of laser light [72]. In addition to detection of exogenous contrast agents (see Section 3), endogenous biomarkers of perfusion and hypoxia can be derived due to the different light absorption spectra of oxy- and deoxyhaemoglobin [73]. In patient-derived head and neck squamous cell carcinoma (HNSCC) xenografts early changes in haemoglobin oxygen saturation following radiotherapy correlated with subsequent changes in tumour volume [74]. Although the MSOT spatial resolution of around 500 μm is inferior to OCT a tissue depth of up to 7 cm is possible [75]. Despite depth limitation both techniques have great potential for non-invasive and endoscopic imaging of a wide range of tumours.

### 4.4. PET Imaging of Perfusion

Several PET tracers have been developed to measure perfusion, of which ^15^O-H_2_O has been the most extensively used. ^15^O-H_2_O is an inert PET tracer that freely diffuses across cell membranes and allows absolute quantification of tumour blood flow with a reproducibility comparable to other imaging modalities [76,77]. High tumour blood flow on ^15^O-H_2_O PET before treatment was predictive of poor response to radiotherapy in head and neck cancer [78]. Unfortunately, the short half-life (2 min) of ^15^O that requires an onsite cyclotron has limited the widespread use of the technique. 

## 5. Hypoxia Imaging

Hypoxia is an important biological determinant of radio-sensitivity and is well characterised having first been recognised in the early part of the 20th century [79]. Dysregulated tumour proliferation and angiogenesis, the latter resulting in the formation of structurally and functionally abnormal neovasculature, combine to increase the distance between cells and a sufficient blood supply resulting in chronic hypoxia and nutrient depletion. The abnormal vasculature is also prone to transient occlusion and hypoperfusion, causing acute, fluctuating hypoxia. Both sources of hypoxia contribute to radio-resistance and the transcriptional regulation of many genes associated with tumour growth and survival, notably hypoxia inducible factor 1 (HIF-1) [80]. This has led to great efforts to minimise tumour hypoxia particularly prior to radiotherapy, with variable levels of clinical success [79]. Imaging modalities that are sensitive to hypoxia could be prognostic and potentially improve outcomes by permitting dose and treatment modification or dose painting. 

### 5.1. PET Imaging of Hypoxia

^18^F-labelled 2-nitroimidazole-based markers have been widely used for PET imaging of hypoxia. In an anoxic environment reduction of the NO_2_ moiety of 2-nitroimidazole by nitroreductases produces highly reactive intermediates which bind to many macromolecules and also undergo glutathione conjugation [81]. ^18^F-fluoromisonidazole (^18^F-FMISO) was the first PET tracer for hypoxic imaging to be developed and has subsequently been the most extensively used, with accumulation having been demonstrated in human glioma, head and neck squamous cell carcinomas (HNSCC), breast, lung and renal tumours. In HNSCC patients, high baseline and ongoing ^18^F-FMISO uptake in the first two weeks of uptake was significantly associated with loco-regional recurrence and was used as a rationale for radiation dose escalation [82,83]. The feasibility of dose painting based on hypoxic and nonhypoxic tumour subvolumes to improve local tumour control has also been demonstrated [84]. Similarly, in locally advanced non-small cell lung cancer, ^18^F-FMISO uptake on baseline scans is strongly associated with poor prognosis. Although dose escalation was possible in this study without excessive toxicity it was not shown to improve outcome [85]. 

A major limitation of ^18^F-FMISO is slow clearance due to its lipophilicity and, to achieve acceptable signal-to-background ratios, static imaging is performed at delayed timepoints (typically up to four hours post-injection). ^18^F-fluoroazomycinarabinoside (^18^F-FAZA) and 3-^18^F-fluoro-2-(4-((2-nitro-1*H*-imidazol-1-yl)methyl)-1*H*-1,2,3,-triazol-1-yl)-propan-1-ol (^18^F-HX4) are hydrophilic molecules and thus have improved pharmacokinetics for PET imaging [86]. The prognostic value and feasibility of dose escalation guided by imaging with ^18^F-FAZA has been shown in HNSCC and NSCLC patients [87,88,89,90,91]. In sarcomas, hypoxia identification with ^18^F-FAZA was associated with radio-resistance and recurrence. However, toxicity limited attempts to combine radiotherapy with sunitinib [92]. An alternative PET tracer for hypoxia imaging is Cu(II)-diacetyl-*bis*(N4-methylthiosemicarbazone (Cu-ATSM) which is believed to be reduced from Cu^2+^ to Cu^1+^ and trapped in hypoxic cells [93]. As a predictive marker, Cu-ATSM has shown potential in cervical, rectal, lung and head and neck cancer [93,94,95,96,97]. 

The inherent sensitivity of PET imaging makes it an attractive modality for hypoxia imaging where detection of relatively small changes in oxygen concentration is required and the initial studies as a prognostic marker have been mostly positive. However, the resolution of clinical PET images is typically around 5 mm, which may lead to a lack of sensitivity when subvoxel hypoxia variation exists and is a potential barrier to dose painting based on hypoxia imaging [93]. It should also be recognised that no imaging modality is sensitive to hypoxia alone and even specific PET tracer uptake is dependent to a certain degree on perfusion, cellularity and other biological variables.

### 5.2. MRI Imaging of Hypoxia

Hypoxia imaging is also possible with MRI because dissolved oxygen and deoxyhaemoglobin are paramagnetic and decrease *T*_1_ and *T*_2_* relaxation. Methods that exploit the effects of these molecules on *T*_1_ relaxation are termed oxygen-enhanced (OE) or tumour oxygenation level-dependent (TOLD) MRI, while imaging of *T*_2_* effects is termed blood oxygenation level-dependent (BOLD) MRI [98]. In a typical study, baseline imaging is performed with the patient breathing room air followed by imaging while the patient inhales oxygen to create arterial hyperoxia with the difference in signal between the two images corresponding to the effect of oxygen inhalation. These assays have been shown to correlate with tumour pO_2_ [98,99,100] and several studies have used the techniques to detect or predict response to radiotherapy. In animal prolactinoma and fibrosarcoma tumour models, BOLD was able to predict growth response after a single radiation dose [101] and the technique has now entered early clinical trials in head and neck cancer patients [102]. OE-MRI is still in preclinical development. Nevertheless, in rats with subcutaneous prostate tumours improved oxygenation of tumours after radiotherapy correlated with response and OE-MRI measurements offered better prognostication than BOLD [103]. Additionally, OE-MRI has also been used in mouse models to differentiate radiation necrosis from glioma [104]. 

## 6. Imaging Changes in Tissue Structure

### 6.1. Diffusion-Weighted MRI

Diffusion-weighted imaging (DWI) is an MRI technique that measures the random, or Brownian, movement of water molecules in tissues that can be quantified. The simplest and most commonly used quantifiable metric used in DWI is the apparent diffusion coefficient (ADC) but more complex models such as VERDICT (vascular, extracellular and restricted diffusion for cytometry in tumours) can extract additional data related to cell size, vascular, intra- and extracellular volume fractions and perfusion effects, which may lead to improved detection of early treatment response [105]. Additionally, numerous imaging techniques have evolved from DWI. For example, diffusion tensor imaging (DTI) and diffusion kurtosis imaging (DKI) can provide information on diffusion directionality and tissue microstructure, respectively [106,107]. Recently, filter-exchange imaging (FEXI) has been used to determine the exchange rate of water across the cell membrane [108,109]. 

In tissues, Brownian motion of water is limited by membranes and macromolecules, giving a lower ADC value (indicating restricted diffusion) for intracellular water than extracellular water [110]. ADC has been shown to have a strong inverse correlation with tumour cellularity in glioma, lung and ovarian tumours, but the relationship is less significant for other tumour types [111]. Following radiotherapy, ADC can transiently decrease due to cellular swelling before increasing due to cell death, the latter being associated with decreasing cellularity and a response to treatment in most studies. At later stages, reductions in ADC can occur due to inflammation and fibrosis [112,113]. Unfortunately, many of these processes coexist, resulting in conflicting effects on diffusion imaging and potentially limiting the early predictive value of the technique. Few studies have looked at the time of assessment with DWI, but in the longitudinal assessment of brain metastases from a range of primary sites treated with whole-brain external beam radiotherapy, the optimal timepoint for prediction of response was after seven fractions on day seven to nine [114]. Similarly, in HNSCC patients an increase in ADC one week after radiotherapy was predictive of response with a sensitivity of 86% and specificity of 83% [115] However, in cervical cancer, although diffusion imaging could detect treatment response upon completion of chemoradiotherapy, reimaging performed in the first two weeks of treatment was unable to differentiate complete, partial and non-responders [116]. In rectal cancer, although DWI alone is not sufficiently accurate for prediction of early response following chemoradiotherapy [117], a combination of DWI, ^18^F-FDG-PET/CT and *T*_2_-weighted volumetry permitted early response prediction with a sensitivity and specificity of 75% and 94%, respectively [118]. In primary glioblastoma, DWI can help to differentiate progression from pseudoprogression (Figure 3) [119]. Recently a combination of twelve multi-parametric imaging features (including ADC) differentiated pseudoprogression from progression with a sensitivity and specificity of 71% and 90%, respectively, versus 100% and 20% for ADC alone [120]. Furthermore, a positive correlation has been reported between necrosis and ADC following treatment and, in most studies, tumour recurrence has generally been found to have a lower ADC than radiation necrosis [121,122,123,124].

### 6.2. Chemical Exchange Saturation Transfer MRI 

The MRI techniques discussed so far all image the protons (^1^H) of water molecules, the abundance of which in biological tissues (60–80 M) facilitates imaging at high temporal and spatial resolution. In the presence of a magnetic field, nuclei with spin (e.g., ^1^H and ^13^C) resonate at a frequency that is partly dependent on the electronic environment of the molecule they are part of, for example, amide protons resonate at a different frequency to water protons. This phenomenon is known as chemical shift and means that MR spectroscopy (MRS) can non-invasively detect the presence and relative concentration of multiple metabolites in vivo [125]. However, the low concentration of these metabolites makes MRS a technique with low temporal and spatial resolution. Chemical exchange saturation transfer (CEST) MRI, described in detail elsewhere [126], is a technique that allows the indirect detection of molecules containing exchangeable protons via attenuation of the water signal. This indirect detection offers greatly enhanced sensitivity, facilitating high-resolution imaging. Amide proton transfer (APT), a CEST technique that detects exchangeable amide protons present in mobile peptides and proteins, has been used to detect the higher concentration of proteins present in tumours corresponding to a higher APT signal than surrounding normal tissue (Figure 4) [127]. In neuro-oncology the technique has promise for the differentiation of progression and radiation necrosis in particular and has already been demonstrated in a study of patients with brain metastases [128].

## 7. Imaging Changes in Metabolism

Aberrant nutrient uptake and subsequent metabolism is a feature of malignant tumours that results from the increased demand for the synthesis of proteins, nucleic acids, fatty acids and other macromolecules required for increased growth and proliferation. Aerobic glycolysis, whereby glucose is reduced to lactate even when oxygen is abundant, was first described by Otto Warburg nearly a century ago and has subsequently been observed in many malignant tumours [129,130]. Following treatment, a decrease in tumour metabolic activity precedes changes in structure and volume, making metabolic imaging attractive for detecting early treatment response [125]. 

### 7.1. Imaging Changes in Glycolysis and TCA Cycle Metabolism

2-(^18^F-fluoro)-2-deoxy-D-glucose (^18^F-FDG) is a glucose analogue that is transported into cells and phosphorylated, trapping the tracer intracellularly and allowing identification of glucose-avid tissues upon subsequent PET imaging. ^18^F-FDG-PET is the most commonly used PET tracer and is used as an adjunct to morphological imaging in the follow-up of many tumours following treatment (Figure 5). Its widespread use and standardisation of acquisition has meant that ^18^F-FDG is currently the only functional imaging technique to be (semi)quantified for use in response evaluation criteria, most notably in PERCIST 1.0 and the EORTC guidelines [131,132]. In HNSCC, the negative predictive value for primary and nodal disease with ^18^F-FDG-PET/CT was 99–100% four months after chemoradiotherapy [133,134]. ^18^F-FDG is also useful in several cancers, for example, nonsmall cell lung cancer, for differentiating recurrence from radiation necrosis [135]. However, attempts to shorten the interval scanning time have produced mixed results [136], with a lack of early response often attributed to inflammation and macrophage infiltration, although the evidence for this mechanistically is limited [137]. 

As discussed earlier, the low concentration of biological metabolites and low sensitivity of NMR limits the temporal and spatial resolution of MRS in vivo. The method of dynamic nuclear polarisation (DNP) of ^13^C-labelled substrates is a technique that can increase the signal-to-noise ratio of ^13^C MR spectroscopy and imaging by >10^4^ in vivo [138]. Hyperpolarised (1-^13^C)pyruvate has been the most widely used substrate due to its high polarisation levels, long polarisation lifetime and its position in the glycolytic pathway. Following injection, hyperpolarised (1-^13^C)pyruvate enters cells via monocarboxylate transporters and, in tumours, is predominantly reduced to lactate by lactate dehydrogenase [139]. Compared to ^18^F-FDG, hyperpolarised (1-^13^C)pyruvate has improved specificity for indicating the Warburg effect and may better differentiate inflammation from tumour progression/recurrence [140]. A reduction of (1-^13^C)lactate production following hyperpolarised (1-^13^C)pyruvate injection after antiandrogen therapy has been observed in a prostate cancer patient [141]. In an orthotopic rat glioma model, a reduction in label flux from (1-^13^C)pyruvate to (1-^13^C)lactate was seen in all animals in the first 96 h after radiotherapy despite increases in tumour size, suggesting that (1-^13^C)pyruvate may be useful to differentiate progression and pseudoprogression (Figure 6) [142].

The other readily translatable hyperpolarised substrate is (1,4-^13^C_2_)fumarate, which is hydrated to malate by fumarase. During cell death an increase in membrane permeability results in leakage of fumarase into the extracellular space and an increased rate of malate production following hyperpolarised (1,4-^13^C_2_)fumarate injection which, in preclinical studies, has been shown to be a sensitive indicator of cell death [143,144]. 

### 7.2. Imaging Proliferation

Several PET tracers have been designed as biomarkers of proliferation. 3′-deoxy-3′-^18^F-fluorothymidine (^18^F-FLT) is taken into cells and phosphorylated by thymidine kinase, the first step of the thymidine salvage pathway essential for DNA synthesis. Thus, ^18^F-FLT preferentially accumulates in cells undergoing proliferation with potentially greater tumour specificity than ^18^F-FDG. Several systematic reviews have concluded that (^18^F-FLT has potential as a marker of early response and shown that a change in uptake correlated well with progression-free and disease-free survival [145,146]. In HNSCC, a comparison of ^18^F-FLT and ^18^F-FDG during radiotherapy showed the overall accuracy of ^18^F-FLT to be significantly higher (74 vs. 30%) [147]. However, following chemoradiotherapy in rectal cancer, despite correlations with disease-free survival, decreases in ^18^F-FLT uptake did not correlate with pathological response, a discrepancy attributed to changes in perfusion following radiotherapy [148].

### 7.3. PET Imaging of Brain Tumours

Lack of tumour specificity of ^18^F-FDG is a particular problem in neuroradiology where there is high background uptake from normal brain tissue and radiation necrosis is often hypermetabolic [149]. Numerous tracers have been designed that are superior to ^18^F-FDG for detection of recurrence and differentiation from pseudoprogression. Brain tumours often have increased uptake of amino acids relative to normal brain and several have been labelled with ^11^C and ^18^F for PET imaging [150]. ^11^C-methionine has been the most widely used amino acid PET tracer. It can differentiate tumours from normal brain with an accuracy of 94% versus 80% for ^18^F-FDG [151] and is also more sensitive for differentiating recurrence from radiation necrosis following radiotherapy [152]. However, the application of ^11^C-labelled substrates will always be limited by the short half-life (20 min) requiring onsite production of the tracer. Therefore, several alternatives have been developed including ^18^F-fluoro-ethyl-tyrosine, an artificial amino acid that is not incorporated into proteins but has increased uptake into tumours [153,154]. Decreased uptake in the first 10 d following chemoradiotherapy was predictive of progression-free survival with an accuracy of 75% [155]. Other tracers that have demonstrated improved performance over ^18^F-FDG for distinguishing recurrence from radiation necrosis include ^11^C- and ^18^F-labelled choline (Figure 4), surrogate measures of the rate of phospholipid membrane synthesis, and ^18^F-dihydroxyphenylalanine (^18^F-DOPA), an analog of the dopamine precursor L-DOPA (Figure 7) [156,157,158].

## 8. Conclusions and Future Perspectives

There has been an explosion in the number of functional imaging techniques that can non-invasively report on multiple biological characteristics of the tumour microenvironment with great potential to guide therapy and improve outcomes as personalised therapy in oncology becomes realistic. Several functional imaging techniques have already been clinically translated, including ^18^F-FDG-PET and DWI-MRI. Detection of early treatment response remains challenging but, as highlighted in this review, there are numerous functional imaging biomarkers that are sensitive to the early biological effects of radiation therapy and can provide prognostic information and guide future treatment. 

There are several common limitations that affect many imaging studies. Most studies are technically challenging and expensive and therefore recruit a small number of patients (typically <50). Quantification is seen as a major strength of functional imaging but a lack of consensus over the vast number of imaging biomarkers to use significantly limits the comparison of findings and meta-analysis. Unfortunately, in clinical practice, quantitative metrics do not necessarily perform better than simple qualitative analysis [159]. Furthermore, few studies prospectively define cutoff points or perform multicentre or external validation and variation between scanners is a significant barrier to quantitative analysis. Reference to the imaging biomarker roadmap should help to address these limitations and facilitate the translation of functional imaging biomarkers into clinical practice [160]. 

## Figures and Tables

**Figure 1 cancers-11-00131-f001:**
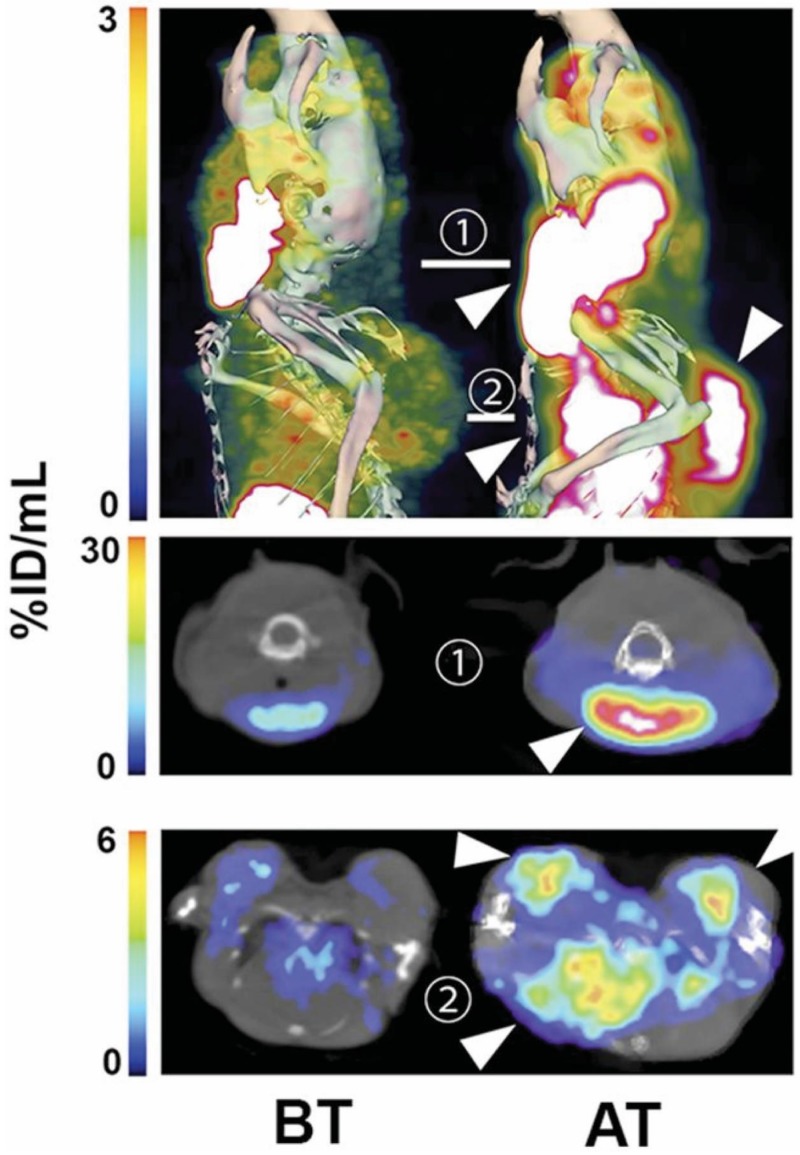
Single photon emission computed tomography (SPECT) imaging of cell death in vivo in Eμ-*Myc* tumours. Imaging of (^99m^Tc)-labelled C2Am at 2 h after probe administration and 24 h after drug treatment. SPECT/CT of representative Eμ-*Myc* mice before (BT, left) and after (AT, right) cyclophosphamide treatment. Top—maximum intensity projections (MIPs); middle—axial section through the cervical tumour; bottom—axial section through the axillary and mediastinal tumours. Tumours are indicated with arrowheads. Reproduced with permission from Neves et al. © SNMMI [20].

**Figure 2 cancers-11-00131-f002:**
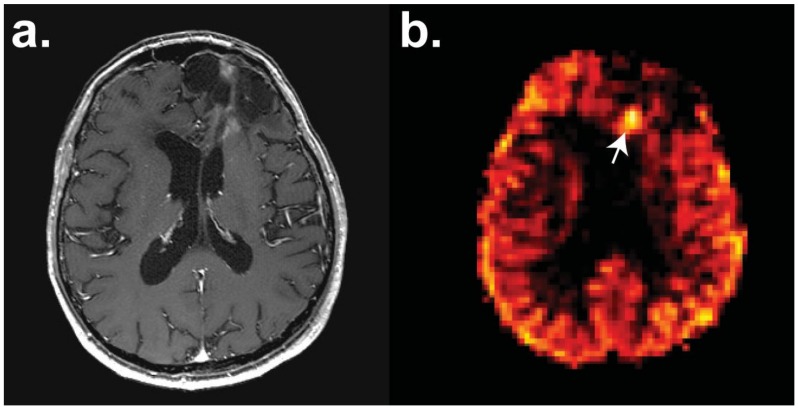
Perfusion magnetic resonance imaging (MRI) with pulsed arterial spin labelling (pASL) of a patient with a frontal high-grade oligodendroglioma with a focus of hyperperfusion (white arrow) identifying recurrent disease post-surgery and chemotherapy. Axial sections have the following sequences: (**a**) *T*_1_-weighted contrast-enhanced and (**b**) pASL. Images courtesy of Dr Harpreet Hyare, University College London Hospital.

**Figure 3 cancers-11-00131-f003:**
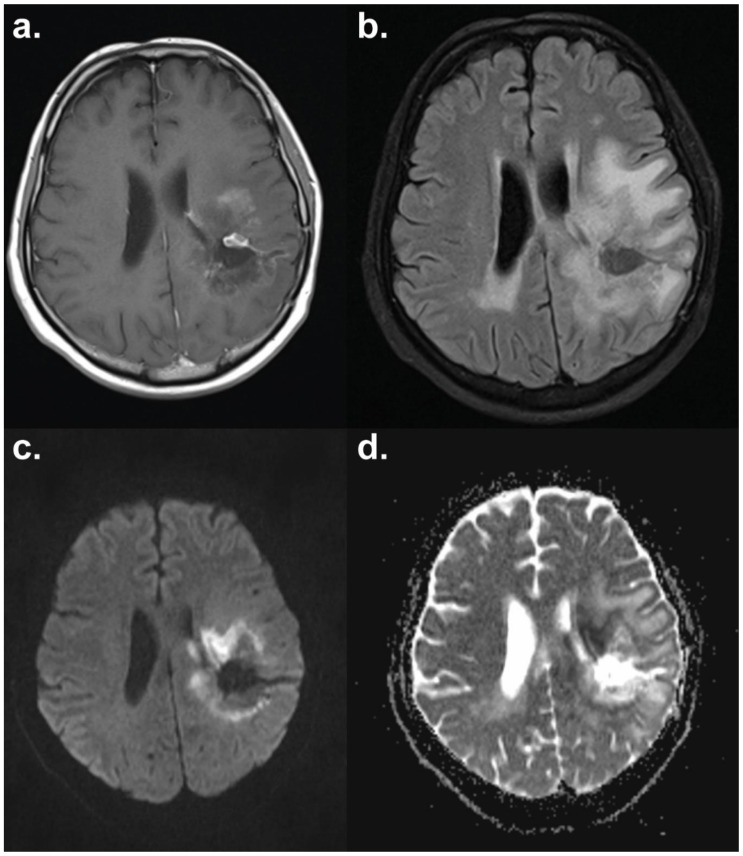
Diffusion-weighted MRI of a patient with glioblastoma multiforme. The necrotic core shows facilitated diffusion with a surrounding ring of restricted diffusion indicative of progressive disease. Axial sections have the following sequences: (**a**) *T*_1_-weighted contrast-enhanced; (**b**) *T*_2_ FLAIR (fluid-attenuated inversion recovery); (**c**) diffusion-weighted imaging (DWI); (**d**) apparent diffusion coefficient (ADC) map. Images courtesy of Dr Harpreet Hyare, University College London Hospital.

**Figure 4 cancers-11-00131-f004:**
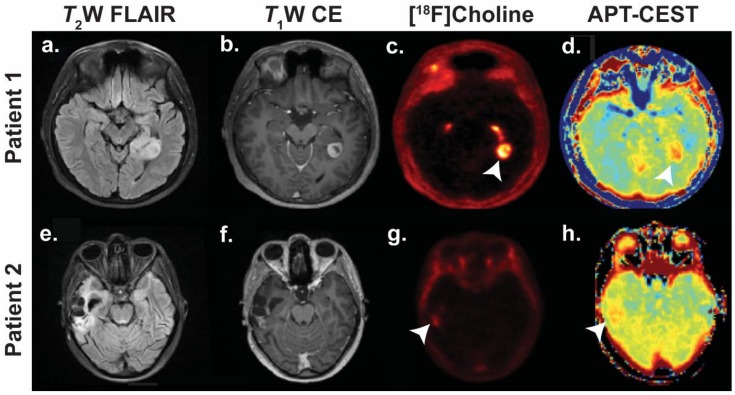
Detection of recurrence in two patients using ^18^F-choline and amide proton transfer- chemical exchange saturation transfer (APT-CEST). Patient 1 (**a**–**d**): WHO grade I pilocytic astrocytoma initially treated with radiotherapy. Imaging performed four years later when the patient re-presented with seizures. Patient 2 (**e**–**h**): WHO grade IV glioblastoma multiforme treated with surgery and chemoradiotherapy. Imaging surveillance performed eighteen months after completing initial treatment. Tumours indicated with white arrowheads. Abbreviations: *T*_2_W FLAIR, *T*_2_-weighted fluid-attenuated inversion recovery; *T*_1_W CE, *T*_1_-weighted contrast enhanced. Images courtesy of Dr Harpreet Hyare, University College London Hospital.

**Figure 5 cancers-11-00131-f005:**
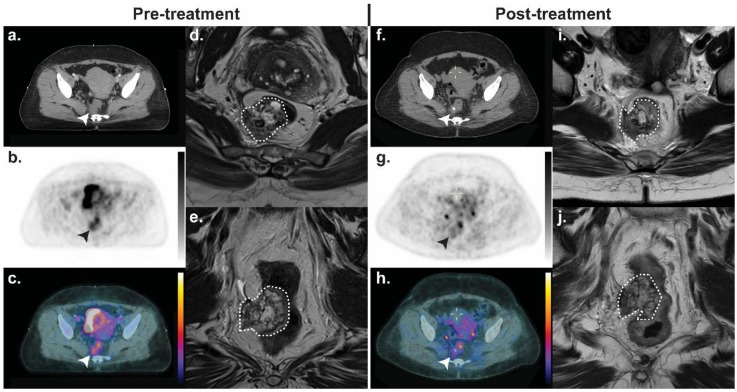
A T4N0M0 rectal adenocarcinoma treated with neoadjuvant chemoradiotherapy. (**b** and **c**) Prior to treatment, the lesion is ^18^F-FDG-avid; (**g** and **h**) after chemoradiation, there is minimal residual uptake despite a minimal change in tumour volume. At subsequent resection, no residual disease remained and the patient was down-staged to ypT0N0M0. (**a** and **f**) axial CT; (**b** and **g**) axial ^18^F-FDG-PET; (**c** and **h**) PET/CT fusion; (**d** and **i**) axial *T*_2_W MRI; (**e** and **j**) coronal *T*_2_W MRI. The tumour is indicated by arrowheads and dashed outline. All ^18^F-FDG-PET images are scaled between an standardised uptake value (SUV) of 0–5.

**Figure 6 cancers-11-00131-f006:**
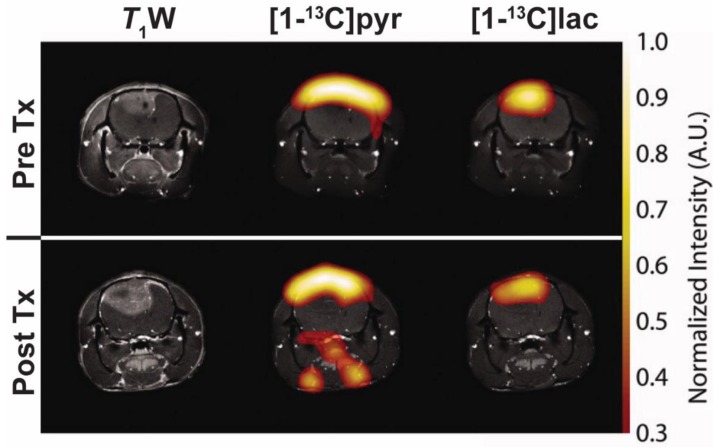
Representative images following injection of hyperpolarised (1-^13^C)pyruvate in a C6 glioma-bearing rat before (top, PreTx) and 96 h after radiotherapy (bottom, PostTx). Images of (1-^13^C)pyruvate ((1-^13^C)pyr) and (1-^13^C)lactate ((1-^13^C)lac) overlaid on coronal *T*_1_W MR images. A reduction in (1-^13^C)lactate production was observed in all tumours after 15 Gy irradiation without decreases in tumour volume. Adapted with permission from Day et al. [142].

**Figure 7 cancers-11-00131-f007:**
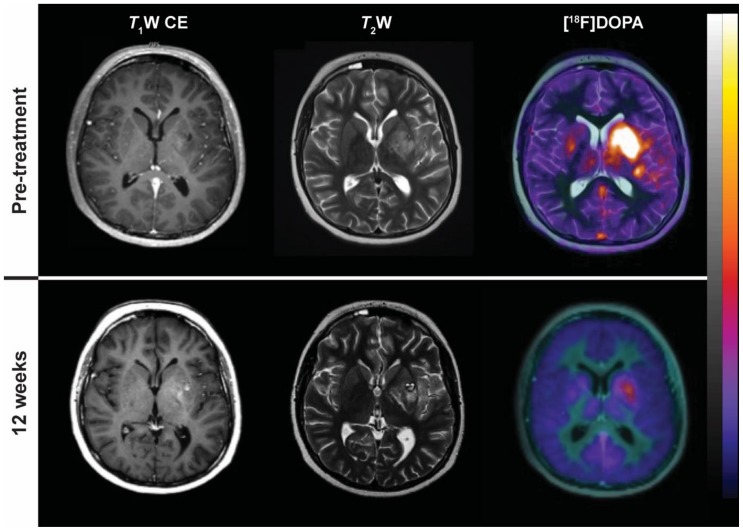
^18^F-dihydroxyphenylalanine (^18^F-DOPA) uptake in a patient with glioblastoma multiforme treated with chemoradiotherapy before (top row) and after 12 weeks (bottom row). From left to right: Post-gadolinium *T*_1_W axial MRI; *T*_2_W axial MRI; fusion PET/MRI with ^18^F-DOPA.

**Table 1 cancers-11-00131-t001:** Comparison between the most frequently used imaging response criteria.

	WHO	RECIST 1.0	RECIST 1.1
Measurement	Sum of maximal perpendicular diameters of all measured lesions	Sum of long axis of up to 10 target lesions	Sum of long axis of up to 5 target lesions (short axis for lymph nodes)
Complete response	Disappearance of all known disease	Disappearance of all target lesions	Disappearance of all target lesions
Partial response	≥50% decrease in lesion size and no new lesions	≥30% decrease in sum of target lesion diameters	≥30% decrease in sum of target lesion diameters
No change/ Stable disease	Neither partial response or progressive disease	Neither partial response or progressive disease	Neither partial response or progressive disease
Progressive disease	≥25% increase in lesion size or ≥1 new lesion	≥20% increase in the sum of target lesions (no minimum size increase)	≥20% increase in the sum of target lesions (≥5 mm absolute increase)
Functional imaging	None	None	^18^F-FDG-PET can be used to complement CT

WHO, World Health Organisation; RECIST, Response evaluation criteria in solid tumours; CT, computed tomography; FDG-PET, positron emission tomography with 2-^18^F-fluoro-2-deoxy-D-glucose.

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
