# Peer review of "Emerging Functional Imaging Biomarkers of Tumour Responses to Radiotherapy"

_cancers, 2019, doi:10.3390/cancers11020131_

Reviewer 1 Report

This is a well written useful straightforward overview of biological targets related to radiotherapy that can be exploited by functional imaging modalities including MRI based and PET based.  The article is broken down by each process (apoptosis, vasculature, hypoxia, tissue structure, metabolism, proliferation.  Each of these sections is interesting and overviews functional imaging approaches for each of these biological processes.

 Nice figures are provided throughout.

 The Introduction initially describes imaging response criteria which provides useful background, then discusses some radiobiology and follows this with a paragraph explaining the focus of the review.  I think this paragraph needs to come earlier in this Introduction as it is not clear why the radiobiology is being explained in that paragraph.  Since these response assessment criteria are discussed it would useful later in the article to discuss any response criteria for functional imaging.

 The article is related to imaging biomarkers related to radiotherapy. More explanation would be useful of why that might be useful and at what timepoint. For example this can be during a course of radiotherapy and could guide and adaptive treatment approach (eg. if a predictive biomarker could be identified) such as escalation or de-escalation of treatment.  Alternatively, imaging can assess response after radiotherapy.  This distinction is important and needs to be elaborated.

 Figure 4 legend needs more explanation about the clinical scenarios eg. what treatment, how long after treatment is imaging.

Author Response

Thank you for your comments on this manuscript. In response we have made the following alterations:

1.     The Introduction initially describes imaging response criteria which provides useful background, then discusses some radiobiology and follows this with a paragraph explaining the focus of the review.  I think this paragraph needs to come earlier in this Introduction as it is not clear why the radiobiology is being explained in that paragraph.  Since these response assessment criteria are discussed it would useful later in the article to discuss any response criteria for functional imaging.

a.     We have changed the structure of the introduction and now include the Biological Effects of Radiation as a seperate section (Section 2).

b.     The only quantification of functional imaging that is in routine clinical use is [18F]FDG SUV. Even then, quantification for response is not that regularly used, with the evidence suggesting that its value over subjective interpretation is often limited. We have adapted Section 7 to reflect the inclusion of FDG in several clinical response guidelines:    "Its widespread use and standardisation of acquisition has meant that [18F]FDG is currently the only functional imaging technique to be (semi)-quantified for use in response evaluation criteria, most notably in PERCIST 1.0 and the EORTC guidelines[131,132]."

2.     The article is related to imaging biomarkers related to radiotherapy. More explanation would be useful of why that might be useful and at what timepoint. For example this can be during a course of radiotherapy and could guide and adaptive treatment approach (eg. if a predictive biomarker could be identified) such as escalation or de-escalation of treatment.  Alternatively, imaging can assess response after radiotherapy.  This distinction is important and needs to be elaborated.

a.     We have adapted the introduction accordingly: "Identification of radio-resistant tumour regions at a pre- or early therapy stage could be used for localised or global dose escalation or initiation of concomitant chemotherapy. Following treatment, functional imaging is a potentially powerful tool where the conclusions of morphological imaging can be limited, for example differentiation of radiation necrosis and residual disease in the brain."

3.     Figure 4 legend needs more explanation about the clinical scenarios eg. what treatment, how long after treatment is imaging.

a.     We currently don't have further clinical details available as Dr Hyare (who provided the images) is on annual leave until mid-January. However, once she returns we are not aware of any reason why we could not expand this legend to include the suggested clinical details before publication.

Reviewer 2 Report

General Comment: This manuscript reviews the current state of functional imaging techniques for non-invasive detection of radiation therapy-induced changes in tumors. It is very well written and presented on an important and timely topic. As a result, it was a pleasure to read and review.

In my opinion, this paper is likely to generate substantial impact and can be published after taking into account a few minor comments below.  

1.      Figure 2: arrow in (b) is not specified in the figure caption.

2.      Section 3 “Imaging changes in microvasculature”, lines 120-122: “Radiation therapy results in acute endothelial cell dysfunction, apoptosis and disruption of blood vessels. Capillaries increase in permeability and become thrombosed due to platelet aggregation and microthrombus formation with subsequent hypo-perfusion causing further tumour necrosis[10,28].”

I would suggest to stress here that vascular damage has been found to have a minimum threshold of single dose of ~8-10Gy of radiation [1-3] due to activation of chemoprotection mechanisms of endothelial cells. This becomes more important in stereotactic radiotherapy where fewer fractions of higher doses are used.

3.      Section 3 “Imaging changes in vasculature” could also cover recent advances in detection of radiation-induced microvascular changes using optical microscopy techniques (e.g., optical coherence tomography [4] and confocal fluorescence microscopy [5]).

References

 1. Garcia-Barros, M. et al. Tumor response to radiotherapy regulated by endothelial cell apoptosis. Science 300, 1155-1159 (2003).

2. Fuks, A., Kolesnick, R. Engaging the vascular component of the tumor response. Cancer Cell 8, 89-91 (2005).

3. Garcia-Barros, M. et al. Impact of stromal sensitivity on radiation response of tumors implanted in scid hosts revisited. Cancer Research 70, 8179-8186 (2010).

4. Demidov, V. et al. Preclinical longitudinal imaging of tumor microvascular radiobiological response with functional optical coherence tomography. Scientific Reports 8(38): 1-12 (2018).

5. Maeda, A. et al. In vivo optical imaging of tumor and microvascular response to ionizing radiation PLoS ONE 7 e42133, 1-15 (2012).

Author Response

Thank you for your comments on this manuscript. In response we have made the following changes:

             1.         Figure 2: arrow in (b) is not specified in the figure caption.

a.     This is now specified in the figure legend."Figure 2. Perfusion MRI with pulsed arterial spin labelling (pASL) of a patient with a frontal high-grade oligodendroglioma with a focus of hyper-perfusion (white arrow) identifying recurrent disease post-surgery and chemotherapy. Axial sections with the following sequences (a) T1-weighted contrast enhanced and (b) pASL. Images courtesy of Dr Harpreet Hyare, University College London Hospital."

2.      Section 3 “Imaging changes in microvasculature”, lines 120-122: “Radiation therapy results in acute endothelial cell dysfunction, apoptosis and disruption of blood vessels. Capillaries increase in permeability and become thrombosed due to platelet aggregation and microthrombus formation with subsequent hypo-perfusion causing further tumour necrosis[10,28].” I would suggest to stress here that vascular damage has been found to have a minimum threshold of single dose of ~8-10Gy of radiation [1-3] due to activation of chemoprotection mechanisms of endothelial cells. This becomes more important in stereotactic radiotherapy where fewer fractions of higher doses are used.

a.     The first paragraph of “Imaging changes in vasculature” has been changed to include reference to threshold dose and the effects of activation of membrane signalling pathways. “Radiation therapy results in acute endothelial cell dysfunction, apoptosis and disruption of blood vessels. Above doses of 8 - 10 Gy endothelial cell apoptosis is induced by activation of the acid sphingomyelinase (ASMase) / ceramide signalling pathway{Garcia-Barros, 2003 #182;Garcia-Barros, 2010 #184;Fuks, 2005 #186}. Therefore, activation of this pathway does not occur with the lower doses delivered in fractionated radiotherapy, only the higher single doses delivered with stereotactic radiotherapy{Corre, 2013 #187}. Capillaries increase in permeability and become thrombosed due to platelet aggregation and microthrombus formation with subsequent hypo-perfusion causing further tumour necrosis{Jonathan, 1999 #35;Miyatake, 2015 #36}. This suggests that imaging changes in perfusion has potential for early detection of tumour responses to radiotherapy.”

3.           Section 3 “Imaging changes in vasculature” could also cover recent advances in detection of radiation-induced microvascular changes using optical microscopy techniques (e.g., optical coherence tomography [4] and confocal fluorescence microscopy [5]).

a.     The contrast enhanced ultrasound section has been renamed “Ultrasound and optical imaging” and now includes a section on OCT and MSOT. “Optical coherence tomography (OCT) is a non-invasive imaging technique that can produce 3D in vivo images at a resolution of a few micrometres by measuring the interference pattern of back scattered light{Zysk, 2007 #190}. Although OCT has unrivalled spatial resolution the scattering of light within biological tissues limits the imaging depth to a few millimetres. OCT is well established for high-resolution 3D retinal imaging and more recently functional imaging of the microvasculature of tumours has been demonstrated using speckle variance OCT{Demidov, 2018 #188}. In pancreatic human tumour xenografts irradiated with ³10 Gy, the vascular volume density decreased by 26% just 30 min post-radiotherapy. Early changes were predominantly seen in small vessels<30 mm in diameter and were transient potentially indicating rapid microthrombus formation following radiotherapy{Maeda, 2012 #189}. Maximal reductions in vascular volume density were seen after two – four weeks, depending on delivered dose, and preceded reductions in tumour volume by several weeks{Demidov, 2018 #188}. Multi-spectral optoacoustic tomography (MSOT) uses an ultrasound transducer to measure acoustic waves generated in response to localised thermoelastic expansion of tissue induced by pulses of laser light{McNally, 2016 #191}. In addition to detection of exogenous contrast agents (see Section 3) endogenous biomarkers of perfusion and hypoxia can be derived due to the different light absorption spectra of oxy- and deoxyhaemoglobin{Tomaszewski, 2018 #192}. In patient derived head and neck squamous cell carcinoma (HNSCC) xenografts early changes in haemoglobin oxygen saturation following radiotherapy correlated with subsequent changes in tumour volume{Rich, 2016 #194}. The MSOT spatial resolution of around 500 mm is inferior to OCT, however a tissue depth of up to 7 cm is possible and both techniques have great potential for non-invasive and endoscopic imaging of a wide range of tumours{Zackrisson, 2014 #193}.”

Reviewer 3 Report

The authors concisely discuss functional imaging techniques and markers for the evaluation of early tumour response to radiotherapy based on biological changes rather than morphological changes. Several functional imaging methods of important hallmarks of cancer and its microenvironment are described. The importance of imaging biomarkers of therapy response and as well as their current limitations are clearly reasoned. The manuscript is well written and well structured, however, some sections could benefit from a more complete overview. For example, 15O-H2O-PET is used in the clinic as a measure of blood volume and vascular permeability, and ADC quantification in DW-MRI correlates with necrosis, providing predictive value in the evaluation of treatment response. Therefore, I suggest the authors add these techniques to the relevant sections of the manuscript. 

Author Response

Thank you for your comments on this manuscript. In response we have made the following changes:

1. The manuscript is well written and well structured, however, some sections could benefit from a more complete overview.

   a. For example, 15O-H2O-PET      is used in the clinic as a measure of blood volume and vascular      permeability.

Response: We have included additional sections in the "Imaging Changes in Vasculature" section:

4.3. Ultrasound and optical imaging

Radiation induced changes in vasculature can also be imaged using dynamic contrast-enhanced ultrasound (CEUS). Low solubility, gas-containing microbubbles have different acoustic properties and can image microvascular density and perfusion. This has been used to predict response of a number of different cancer types to chemotherapy[63-65], while decreased vascular density following radiation therapy has been used as an early marker of response in pre-clinical tumour models[66,67]. Furthermore, by using antibodies conjugated to the surface of microbubbles could permit tumour targeting. Microbubbles targeting the angiogenesis regulators αvβ3 integrin and ICAM-1 that are up-regulated in response to radiotherapy were increased in a rat prostate tumour model so treated[68].

Optical coherence tomography (OCT) is a non-invasive imaging technique that can produce 3D in vivo images at a resolution of a few micrometres by measuring the interference pattern of back scattered light[69]. Although OCT has unrivalled spatial resolution the scattering of light within biological tissues limits the imaging depth to a few millimetres. OCT is well established for high-resolution 3D retinal imaging and more recently functional imaging of the microvasculature of tumours has been demonstrated using speckle variance OCT[70]. In pancreatic human tumour xenografts irradiated with ³10 Gy, the vascular volume density decreased by 26% just 30 min post-radiotherapy. Early changes were predominantly seen in small vessels<30 mm in diameter and were transient potentially indicating rapid microthrombus formation following radiotherapy[71]. Maximal reductions in vascular volume density were seen after two – four weeks, depending on delivered dose, and preceded reductions in tumour volume by several weeks[70].

Multi-spectral optoacoustic tomography (MSOT) uses an ultrasound transducer to measure acoustic waves generated in response to localised thermoelastic expansion of tissue induced by pulses of laser light[72]. In addition to detection of exogenous contrast agents (see Section 3), endogenous biomarkers of perfusion and hypoxia can be derived due to the different light absorption spectra of oxy- and deoxyhaemoglobin[73]. In patient derived head and neck squamous cell carcinoma (HNSCC) xenografts early changes in haemoglobin oxygen saturation following radiotherapy correlated with subsequent changes in tumour volume[74]. The MSOT spatial resolution of around 500 mm is inferior to OCT, however a tissue depth of up to 7 cm is possible[75]. Nevertheless, both techniques have great potential for non-invasive and endoscopic imaging of a wide range of tumours.

4.4. PET imaging of perfusion

     Several PET tracers have been developed to measure perfusion of which [15O]H2O has been the most extensively used. [15O]H2O is an inert PET tracer that freely diffuses across cell membranes and allows absolute quantification of tumour blood flow with a reproducibility comparable to other imaging modalities[76,77]. High tumour blood flow on [15O]H2O PET before treatment was predictive of poor response to radiotherapy in head and neck cancer[78]. Unfortunately, the short half-life (2 min) of 15O that requires an on-site cyclotron has limited the widespread use of the technique.

    b. ADC quantification in DW-MRI correlates with necrosis, providing predictive value in the evaluation of treatment response.

Response: We have amended the DWI section to better reflect the role of ADC as a biomarker of cellularity and necrosis. Specifically:

In tissues Brownian motion of water is limited by membranes and macromolecules, giving a lower ADC value (indicating restricted diffusion) for intracellular water than extracellular water[110]. ADC has been shown to have a strong inverse correlation with tumour cellularity in glioma, lung and ovarian tumours, but the relationship is less significant for other tumour types[111].

In primary glioblastoma DWI can help to differentiate progression from pseudoprogression (Figure 3)[119]. Recently a combination of twelve multi-parametric imaging features (including ADC) differentiated pseudoprogression from progression with a sensitivity and specificity of 71% and 90% vs. 100% and 20% for ADC alone[120]. Furthermore, a positive correlation has been reported between necrosis and ADC following treatment and in most studies tumour recurrence has generally been found to have a lower ADC than radiation necrosis[121-124].